# Transcriptomic Analyses and Experimental Validation Identified Immune-Related lncRNA–mRNA Pair *MIR210HG*–*BPIFC* Regulating the Progression of Hypertrophic Cardiomyopathy

**DOI:** 10.3390/ijms25052816

**Published:** 2024-02-29

**Authors:** Yuan Zhang, Jiuxiao Zhao, Qiao Jin, Lenan Zhuang

**Affiliations:** 1Institute of Genetics and Reproduction, College of Animal Sciences, Zhejiang University, Hangzhou 310058, China; zhangyuan2020@zju.edu.cn (Y.Z.); 3190100251@zju.edu.cn (J.Z.); 22117037@zju.edu.cn (Q.J.); 2Key Laboratory of Cardiovascular Intervention and Regenerative Medicine of Zhejiang Province, Department of Cardiology, Sir Run Run Shaw Hospital, College of Medicine, Zhejiang University, Hangzhou 310016, China

**Keywords:** hypertrophic cardiomyopathy, bioinformatics analysis, immune cell infiltration, non-coding RNAs

## Abstract

Hypertrophic cardiomyopathy (HCM) is a disease in which the myocardium of the heart becomes asymmetrically thickened, malformed, disordered, and loses its normal structure and function. Recent studies have demonstrated the significant involvement of inflammatory responses in HCM. However, the precise role of immune-related long non-coding RNAs (lncRNAs) in the pathogenesis of HCM remains unclear. In this study, we performed a comprehensive analysis of immune-related lncRNAs in HCM. First, transcriptomic RNA-Seq data from both HCM patients and healthy individuals (GSE180313) were reanalyzed thoroughly. Key HCM-related modules were identified using weighted gene co-expression network analysis (WGCNA). A screening for immune-related lncRNAs was conducted within the key modules using immune-related mRNA co-expression analysis. Based on lncRNA–mRNA pairs that exhibit shared regulatory microRNAs (miRNAs), we constructed a competing endogenous RNA (ceRNA) network, comprising 9 lncRNAs and 17 mRNAs that were significantly correlated. Among the 26 lncRNA–mRNA pairs, only the *MIR210HG–BPIFC* pair was verified by another HCM dataset (GSE130036) and the isoprenaline (ISO)-induced HCM cell model. Furthermore, knockdown of *MIR210HG* increased the regulatory miRNAs and decreased the mRNA expression of *BPIFC* correspondingly in AC16 cells. Additionally, the analysis of immune cell infiltration indicated that the *MIR210HG*–*BPIFC* pair was potentially involved in the infiltration of naïve CD4^+^ T cells and CD8^+^ T cells. Together, our findings indicate that the decreased expression of the lncRNA–mRNA pair *MIR210HG*–*BPIFC* was significantly correlated with the pathogenesis of the disease and may be involved in the immune cell infiltration in the mechanism of HCM.

## 1. Introduction

Hypertrophic cardiomyopathy (HCM) is a disease induced by multiple conditions and characterized by unexplained, isolated, and progressive cardiac hypertrophy [1]. Epidemiological investigations estimate the population prevalence of HCM to be approximately 0.2%, with this disease being the leading cause of sudden cardiac death among young adults and athletes [2]. Patients with HCM display significant clinical variations, with left ventricular hypertrophy, myofibrillar anomalies, and fibrosis being the predominant manifestations. Myocardial hypercontractility frequently coexists with heart failure or arrhythmia. The current therapeutic approaches for HCM include interventional surgery and drug therapy, both of which have demonstrated a certain efficacy [3]. However, no treatment has been applied to reversing it in clinical practice, thus requiring the establishment of new treatment strategies. Extensive investigations have consistently shown a compelling association between infrequent immune activation, immune cell infiltration [4], and the occurrence of cardiovascular diseases, most notably atherosclerosis, coronary heart disease, and heart failure [5,6]. At the same time, the immune response is one of the important reasons for the development of HCM. Numerous studies have shown that HCM patients have leukocyte infiltration and elevated levels of inflammatory cytokines in their myocardium, which may be key factors in the progression of HCM to the end stage of heart failure [7]. In addition, the genetic deficiency of the key inflammatory cytokine IL-6 has been shown to alleviate TAC-induced left ventricular dysfunction and hypertrophy [8]. IL-6, which is a typical cytokine associated with inflammation, can also induce the differentiation of naïve T cells into CD8^+^ T cells [9]. These studies collectively demonstrate the crucial role of the immune system in the pathophysiology of left ventricular hypertrophy.

Long non-coding RNAs (lncRNAs) are RNA molecules with transcripts longer than 200 nucleotides that lack protein-coding ability. These lncRNAs competitively interact with microRNAs (miRNAs) to hinder the degradation of target mRNA via competitive endogenous RNA (ceRNA) regulatory mechanisms, thereby participating in various fundamental biological processes [10]. Recent studies have revealed the crucial role of lncRNAs in the progression of several cardiovascular conditions, such as myocardial fibrosis, myocardial hypertrophy, and atherosclerosis [11]. Several lncRNAs, including *PDIA3P*, *TSPYL3*, *LOH3CR2A*, *LOC401431*, *LOC158376*, and *LOC606724*, have been linked to the pathogenesis of HCM [12]. Although some reports have linked specific lncRNAs to HCM, the lack of genome-wide analyses and verification means that the role of immune-related lncRNA–mRNA pairs in the progression of HCM remains unclear.

The rapid progression of whole-transcriptome analysis has led to the widespread utilization of bioinformatics analyses in predicting disease-associated genes, elucidating the underlying mechanisms of diseases, and exploring potential therapeutic targets [13]. In this study, we obtained the transcriptomic profiles of mRNAs and lncRNAs from patients diagnosed with HCM and healthy controls by accessing publicly available RNA-Seq data. We employed weighted gene co-expression network analysis (WGCNA) to construct a correlation network based on the training dataset, enabling us to identify two modules associated with HCM. Additionally, we identified lncRNAs associated with the immune system within the module, and we constructed a ceRNA network consisting of two immune-associated lncRNAs and four immune-associated mRNAs, which were reciprocally regulated by shared miRNAs. The correlation of the lncRNA–mRNA pair *MIR210HG*–*BPIFC* was confirmed through verification using the validating dataset and experiments in an HCM cell model. Knockdown of the lncRNA *MIR210HG* by siRNA decreased the mRNA levels of *BPIFC*, with more expression of the regulatory miRNAs and hypertrophy, which further confirmed the important role of the *MIR210HG*–*BPIFC* pair in the pathogenesis mechanism of HCM. The analysis of immune cell infiltration indicated that the lncRNA–mRNA pair *MIR210HG*–*BPIFC* may be involved in the infiltration of naïve CD4^+^ T cells and CD8^+^ T cells in HCM. Together, our findings identified the immune-related lncRNA–mRNA pair *MIR210HG*–*BPIFC* regulating the progression of HCM.

## 2. Results

### 2.1. Data Collection and Analysis of Differentially Expressed Genes

To investigate the potential alterations in lncRNA–mRNA pairs within the cardiac myocytes of patients with hypertrophic cardiomyopathy (HCM), we chose the RNA-Seq dataset GSE180313, which consists of heart samples obtained from 13 HCM patients and 7 healthy controls (Appendix A). Compared to other RNA-Seq datasets for patients with HCM, the patient samples selected in GSE180313 were identified based on histological examination in addition to imaging, which made their HCM patient samples more reliable. We conducted cluster tree analysis on the complete transcriptome data (Appendix A), and the results indicated that there were no samples that exhibited any specific characteristics requiring exclusion. Then, we identified differentially expressed genes (DEGs) following specific criteria (|Log2 fold change| > 1 and adjusted *p*-value < 0.05). The analysis identified a total of 507 upregulated genes and 191 downregulated genes (Appendix A). Figure 1A,B display the heatmap and the volcano plot of the DEGs, respectively. These results show that the GSE180313 dataset is reliable.

### 2.2. Construction of Weighted Co-Expression Network and Identification of HCM-Related Key Modules

Most biological networks are scale-free networks, which exhibit a power-law distribution. These distribution characteristics render biological networks both robust and fragile. WGCNA employs correlation coefficient values to construct a gene co-expression network that adheres to a scale-free network distribution. Thus, to identify crucial lncRNA–mRNA pairs, we utilized the GSE180313 dataset to build a WGCNA network. Since the number of DEGs was obviously not enough to build a robust network, we chose the top 5000 genes with the highest mean absolute deviation (MAD) in the GSE180313 dataset. The pickSoftThreshold function from the WGCNA package was employed to confirm β = 5 (Figure 2A) as the soft threshold. Subsequently, the WGCNA network was constructed by implementing hierarchical clustering and dynamic tree-cutting functions, resulting in 17 modules, each composed of over 50 genes (Figure 2B). To identify the modules associated with clinical features, we calculated the correlation between each module and the clinical features, and we generated a heatmap (Figure 2C). The heatmap reveals that the yellow module exhibits the strongest negative correlation with HCM, while the turquoise module demonstrates the strongest positive correlation with HCM. The gene expression patterns in the yellow and turquoise modules are also visualized in the heatmaps (Figure 2D,E, respectively). 

### 2.3. Construction of Immune-Related ceRNA Network

Previous studies have demonstrated that HCM patients experience inflammatory cell infiltration and other immune responses in their myocardium. However, the precise mechanism through which the immune system contributes to the pathogenesis of HCM requires additional investigation [14]. Therefore, we conducted further study into the immune-related genes within pivotal modules. A list of immune-related genes was retrieved from the Immunology Database and Analysis Portal (ImmPort), serving as a reference for screening immune-related mRNA expression (Appendix A) and immune-related lncRNA expression (Appendix A) in the turquoise and yellow modules. 

Our hypothesis posits that these immune-related lncRNA molecules function as ceRNAs to exert their regulatory influence. In order to validate this hypothesis, we constructed a ceRNA network incorporating the correlation levels between lncRNA and mRNA, as well as the predicted and experimentally confirmed miRNA–mRNA/lncRNA interactions. Data regarding both predicted and experimentally validated miRNA–lncRNA and miRNA–mRNA interactions were gathered from the miRcode database. The selection of lncRNA–mRNA pairs for constructing the ceRNA network had to satisfy the following criteria: (1) the expression of lncRNA and mRNA must exhibit positive correlation (Pearson’s correlation coefficient > 0.75); (2) the lncRNA and mRNA must share at least one miRNA; (3) there should be differential expression of all lncRNAs and mRNAs between HCM and CTRL.

Based on these assumptions, we initially computed the correlation coefficient between immune-related lncRNAs and immune-related mRNAs. By using a cutoff value of 0.75, we identified 341 potential candidate lncRNA–mRNA pairs, comprising 41 mRNAs and 101 lncRNAs. Subsequently, we assessed whether these pairs shared miRNAs, obtaining 26 lncRNA–mRNA pairs (Table 1), which encompassed 9 lncRNAs and 17 mRNAs. Then, we used Cytoscape to construct a preliminary network diagram (Figure 3A). Lastly, we evaluated whether there were significant differences in the expression levels of these lncRNAs and mRNAs between the HCM group and the CTRL group, using the Kruskal–Wallis test (Figure 3B,C). We discovered that two lncRNAs and five mRNAs exhibited no significant variation in expression between the two groups. Consequently, we excluded these pairs from the analysis, resulting in the construction of a ceRNA network, composed of 16 lncRNA–mRNA pairs. Detailed information on the interactions between lncRNAs, miRNAs, and mRNAs within the network is shown in Appendix A.

### 2.4. Validation of the Immune-Related lncRNA–mRNA Pairs

To validate the ceRNA network obtained from the above analysis, we downloaded a new HCM dataset as a validation dataset, GSE130036, which consists of 28 samples from HCM patients and 9 samples from healthy controls (Appendix A). Initially, we examined whether there were disparities in the expression levels of mRNAs and lncRNAs between the HCM and CTRL samples in the GSE130036 dataset (Figure 3D,E). We found that the lncRNAs *KTN1*-*AS1* and *TMEM72*-*AS1* and the mRNAs *IL33* and *GZMB* showed differential expression levels in GSE180313, while their expression levels did not show significant differences in GSE130036. Consequently, pairs containing them were excluded, and five lncRNA–mRNA pairs remained. Subsequently, we conducted a correlation test on the lncRNA–mRNA pairs (Figure 4A–C). Since the *LINC00092*–*PPP4C* pair’s correlation *p*-value was non-significant (*p*-value = 0.22) in GSE130036 (Figure 4B), this pair was removed. The remaining four lncRNA–mRNA pairs (*ZNF503-AS2–JAK2, ZNF503-AS2–NPR3, ZNF503-AS2–BRAF*, and *MIR210HG–BPIFC*) and related miRNAs were used for constructing the ceRNA network (Figure 4D).

To further validate these four lncRNA–mRNA pairs’ expression correlation, the dysregulation of the four lncRNA–mRNA pairs was further validated in a cardiac hypertrophy cell model [15]. We stimulated AC16 cells with 10 µM isoprenaline (ISO) for 24 h to induce hypertrophy in vitro. The results of actin-tracker staining (Figure 5A) and the mRNA expression of hypertrophy-related genes *Nppa* and *Nppb* (Figure 5B) revealed that the in vitro cardiomyocyte hypertrophy model was successfully established. Then, we detected the expression of four lncRNA–mRNA pairs by qRT-PCR. The results showed that the expression of all four pairs was correspondingly decreased (Figure 5C,D). In contrast, the expression of the pairs including the lncRNA *ZNF503-AS2* was increased in the training dataset and the validation dataset. So, only the pair *MIR210HG*–*BPIFC* was consistently correspondingly decreased in all analyses and experimental validation. Furthermore, we detected the *MIR210HG*–*BPIFC*-related miRNA levels in a cardiomyocyte hypertrophy model. The results showed that *miR-216b*, *miR-24*, *miR-34c*, etc., were increased by ISO stimulation (Figure 5E and Appendix A). Additionally, to further confirm the important role of *MIR210HG*–*BPIFC* in hypertrophy, the *MIR210HG* was knocked down in AC16 cells by two siRNAs (siRNA-1 and siRNA-2). The qRT-PCR results showed that the mRNA level of *BPIFC* was correspondingly decreased with the knockdown of *MIR210HG* (Figure 5F). At the same time, the expression of the mediating miRNAs (*miR-145*, *miR-216b*, *miR-24*, *miR-34c*, etc.) increased compared with the siRNA-NC (Figure 5G and Appendix A). Interestingly, we found that *MIR210HG* knockdown increased the expression levels of *Nppa* and *Nppb* in mRNA (Figure 5H) and induced hypertrophic morphology in AC16 cells (Figure 5I). Therefore, *MIR210HG* is not only a biomarker of HCM, but also regulates the expression of *BPIFC* in *MIR210HG–BPIFC* pairs through unique miRNA and plays an important role in the pathogenesis of cardiomyocyte hypertrophy. 

### 2.5. Analysis of Immune Cell Infiltration

CIBERSORTx is a powerful tool that provides a detailed description of tissue composition using RNA-Seq data and is commonly used for immune cell infiltration analysis [16,17]. To examine the composition of immune cells in HCM and explore the association between immune cell composition and the online CIBERSORTx website, we used the reference expression matrix (LM22) generated from scRNA-Seq of 22 immune-cell types isolated from peripheral blood to calculate the proportions of each immune-cell type in the GSE180313 dataset (Figure 6A). The results of the CIBERSORTx calculation showed a significant increase in CD8^+^ T cells (*p*-value = 0.02392), along with significant decreases in naïve CD4^+^ T cells (*p*-value = 0.0194) and resting mast cells (*p*-value = 0.0194), in the HCM samples of the GSE180313 dataset (Figure 6B). The differences in the cell infiltration of mast cells and T cells were consistent with the observations in previous studies of HCM [18]. 

To investigate the role of the validated lncRNA–mRNA pairs in the immune cell infiltration, we performed correlation analysis between the expression of four lncRNA–mRNA pairs (*ZNF503-AS2–JAK2*, *ZNF503-AS2–NPR3*, *ZNF503-AS2–BRAF*, and *MIR210HG–BPIFC*) and the proportion of immune cell infiltration. The expression of *MIR210HG*–*BPIFC* showed a positive correlation with the infiltration of naïve CD4^+^ T cells and a negative correlation with the infiltration of CD8^+^ T cells (Figure 6C,D). None of the other three pairs were associated with changes in immune cell infiltration (Appendix A). The analysis results suggest that the *MIR210HG–BPIFC* pair may regulate the progression of HCM by affecting the composition of immune cell infiltration.

## 3. Discussion

In our study, we employed an integrated bioinformatics approach to analyze clinical HCM RNA-Seq data. We identified an lncRNA–mRNA pair, *MIR210HG*–*BPIFC*, which potentially regulates the occurrence and development of HCM, and may be involved in the infiltration of naïve CD4^+^ T cells and CD8^+^ T cells. These findings highlight their relevance with HCM, partially through the ceRNA network and immune cell infiltration (Figure 7).

Previous studies have examined lncRNA transcription in the heart tissues of HCM patients using microarray technology and RNA sequencing [19,20,21]. Some lncRNAs have been identified as biomarkers for further analysis. The RNA-Seq dataset GSE180313 contains heart tissues from 7 control individuals and 13 HCM patients. Additionally, due to the small number of DEGs and the fact that they did not conform to the assumptions of WGCNA, we selected the top 5000 genes based on MAD obtained from GSE180313. We constructed gene modules through unsupervised clustering and selected the modules most relevant to HCM. Notably, key modules included genes such as *RHOA* [22] and *JAK2* [23], which have been previously reported to be associated with HCM. Previous studies have confirmed that transcriptional regulation of *JAK2* in verified cases of HCM is correlated with increased global ventriculus sinister and cardiomyocyte nuclear *JAK2* expression, as well as the activation of its downstream canonical target, *STAT3* [24]. Also, an increasing number of signal transduction pathways, including *RHOA*, have been recognized as crucial regulators of the hypertrophic response. *BRAF* belongs to a small family of serine/threonine kinases that act as *RAS* effectors. All three *RAF* family members, namely, *RAF-1*, *BRAF*, and *ARAF*, are expressed in cardiac cells and promote the survival and growth of cardiomyocytes [25]. *Interleukin 33* (*IL-33*) has been confirmed as an emerging immunological and cardiovascular marker [26], which is also included in the key modules. Subsequently, we constructed the immune-related ceRNA network for genes in key modules, based on validated predictions of miRNA–mRNA/lncRNA regulation and their associated levels. In this ceRNA network, both the lncRNA and mRNA must share at least one miRNA, and their expression must exhibit a Pearson’s correlation coefficient greater than 0.75. We identified 26 lncRNA–mRNA pairs through the analysis. Then, 22 pairs were excluded by correlation tests and expression difference tests, and 4 pairs were retained: *ZNF503-AS2–JAK2*, *ZNF503-AS2–NPR3*, *ZNF503-AS2–BRAF*, and *MIR210HG–BPIFC*.

Two candidate lncRNAs in the ceRNA network, namely, *MIR210HG*, and *ZNF503-AS2*, were found to be related to HCM for the first time. The lncRNA *MIR210HG*, the host gene encoding *miR-210*, is located on chromosome 21q13.3 and consists of 567 nucleotides [27]. Previous studies have shown that *MIR210HG* acts as an oncogene and promotes tumor progression. In endometrial cancer, *MIR210HG* enriched genes in the *Wnt* and *TGF-β/Smad3* signaling pathways, thereby enhancing the development of cancer [28]. Similarly, *MIR210HG* is involved in the Warburg effect of inducing tumor growth in breast cancer [29]. We measured the expression of the tumor-related *MIR210HG*-linked miRNAs and mRNAs in an ISO-induced HCM cell model. There were no significant changes in the expression of the miRNAs *miR-608*, *miR-503-5p*, and *miR-337-3p* or the mRNAs *FXO6*, *TRAF4*, and *HMGA2*, related to hepatoblastoma, cervical cancer, and endometrial cancer (Appendix A). These data suggest that the lncRNA–mRNA pair *MIR210HG–BPIFC* and its regulatory miRNAs play specific roles in HCM, different from their roles in tumor conditions. Additionally, Wu et al. found that *ZNF503-AS2* could be used as an independent prognostic biomarker for rhabdoid tumors of the kidney through univariate and multivariate Cox analyses [30]. Furthermore, prognostic and diagnostic models for kidney renal clear-cell carcinoma suggested that *ZNF503-AS2* could be used as a prognostic and diagnostic biomarker in patients [31], since we found that the expression levels of this candidate lncRNA in the ceRNA network were significantly dysregulated in HCM patients. In addition, co-expression analysis showed that these lncRNAs were strongly associated with immune-related genes such as *JAK2*, *NPR3*, *BRAF*, and *BPIFC*. We speculate that these two candidate lncRNAs affect immune-related genes through miRNAs, subsequently impacting immune cells and, ultimately, contributing to the development and progression of HCM.

Through WGCNA and co-expression analysis, several immune-associated lncRNA–mRNA pairs were identified in HCM [13]. However, only one pair, *MIR210HG–BPIFC*, was verified using an independent dataset and an in vitro model of myocardial hypertrophy in AC16 cells. Although the differences between HCM patients and the in vitro cell model may stem from different physiological states, we believe that the *MIR210HG–BPIFC* pair is more significant and accurate. *BPIFC*, a protein-coding gene located on Chr 22q13, is rarely expressed and is involved in lipid transfer and lipopolysaccharide binding [32]. Notably, it shows high expression levels in skin samples from psoriasis patients, and its expression is abnormally elevated in inflamed psoriatic skin, suggesting its involvement in inflammation and/or immune response [33]. Additionally, research has demonstrated the functional roles of *MIR210HG* in various diseases through its interactions with different miRNAs [29]. Therefore, it is plausible that *MIR210HG–BPIFC* is an important immune-associated lncRNA–mRNA pair identified in relation to HCM. Furthermore, we confirmed the downregulation of the *MIR210HG–BPIFC* pair in an ISO-induced hypertrophic cardiomyocyte model. Interestingly, without ISO stimulation, knockdown of *MIR210HG* by siRNA not only decreased the expression of *BPIFC*, but also induced the hypertrophy marker genes and cellular hypertrophic morphology. At the same time, the regulatory miRNAs *miR-145*, *miR-216b*, *miR-24*, *miR-34c*, etc., showed moderately increased expression. These cellular experiments further confirmed the correlation of *MIR210HG–BPIFC* with HCM and indicated the important role of *MIR210HG–BPIFC* in regulating the progression of HCM. However, the confirmation of their association was solely based on an in vitro cardiomyocyte hypertrophy model. This study focused on the correlation between lncRNA and mRNA, and on the association between lncRNA–mRNA pairs and HCM, thus requiring further in vivo studies to validate the specific intervention mechanism.

We further tested our hypothesis by immune cell infiltration. Lots of studies have demonstrated an increase in inflammatory cytokines and immune cell infiltration in the myocardium of individuals with HCM. These findings imply that immune cells significantly contribute to the development of HCM. Previous studies have focused on mast cells. Mast cells were found to produce growth factors such as TGF-β and bFGF [34], along with neurotransmitters such as histamine that trigger positive cardiac effects [35]. All of these factors are closely related to the pathogenesis of HCM. The mast cell stabilizer was able to reduce compulsive left ventricular remodeling in a rat model [36]. A significant downregulation of resting mast cells was also found in our immune cell infiltration analysis. Interestingly, we also found significant decreased infiltration of naïve CD4^+^ T cells and significantly increased infiltration of CD8^+^ T cells, which is consistent with the conclusions of previous studies [37]. Meanwhile, in the published case reports, there have been cases of T-cell lymphoma complicated with HCM [37] and HCM cases after cell therapy intervention with T cells, indicating that T cells also play an important role in the pathogenesis of HCM. T-cell infiltration has been found in the published HCM heart single-cell data [38]. In the cluster of T cells, the proportion of the subgroups with increased expression of CD8^+^ marker genes was increased, indicating that the infiltration of CD8^+^ T cells was increased in HCM [38]. The expression changes of the *MIR210HG–BPIFC* pair were correlated with the immune-related miRNAs in the AC16 model of cardiomyocyte hypertrophy (Figure 5E and Appendix A). Based on our bioinformatics analysis and experimental data, we speculated that the decreased expression of the *MIR210HG–BPIFC* pair in AC16 cells may regulate the innate immune response cytokines and affect the recruitment of naïve CD4^+^ T cells and CD8^+^ T cells. In our study, the lncRNA–mRNA pair *MIR210HG–BPIFC* was found to be highly positively correlated with naïve CD4^+^ T cells and negatively correlated with CD8^+^ T cells. These results suggest that the inhibition of the *MIR210HG–BPIFC* pair may be one of the reasons for the significantly increased infiltration of CD8^+^ cells in HCM. Our data suggest that the *MIR210HG–BPIFC* pair is a potential biomarker for immune cell disorder in HCM. However, the detailed molecular mechanisms are still obscured and need further studies.

However, there are certain limitations to this study. Enhancing the availability of clinicopathological genetic data and increasing the sample size could improve the accuracy of disease assessment and prediction. Additionally, it is important to note that although we validated our predicted results using the AC16 cell model, such cells do not fully represent human beating cardiomyocytes. Therefore, further investigation using clinical samples of human cardiomyocytes and myocardium affected by hypertrophic cardiomyopathy will be necessary to validate the regulatory impact of immune-related lncRNA–mRNA interactions in HCM.

## 4. Materials and Methods

### 4.1. Data Collection

The RNA-Seq datasets GSE180313 (training dataset) and GSE130036 (validation dataset) were downloaded from the Gene Expression Omnibus (GEO) (http://www.ncbi.nlm.nih.gov/geo/ (accessed on 26 October 2021)). GSE180313 consists of heart tissues belonging to 13 HCM patients and 7 normal controls. GSE130036 also includes heart tissues from 28 HCM patients and 9 healthy donors. We downloaded the raw counts, in the fastq.gz file format, of 41 HCM heart tissues and 16 healthy controls from these two databases’ expression profiling for further analysis. More details about the datasets are shown in Appendix A, and Figure 8 illustrates the bioinformatics analysis workflow in our study.

### 4.2. Data Preprocessing and DEG Screening

We used hisat2 (versions 2.1.0-4) to align multiple sets of paired-end fastq files to the reference genome (GRCh37), sort the alignments, and convert them to the BAM format. Then, the BAM format was used for quantifying gene expression by featureCounts (v2.0.2). GSE180313 was chosen as the training database to identify important genes, because it was more credible, while the GSE130036 database served as a validation set. In order to identify the DEGs, we used the “DESeq2” package in R software (version 4.3.1) to filter out genes with a total count of less than 1 in the gene expression matrix, and then we performed differential analysis using the DESeq function. Finally, we identified the DEGs with the cutoff |Log2 fold change| > 1 and adjusted *p*-value < 0.05.

### 4.3. Weighted Gene Co-Expression Network Analysis

The construction of the co-expression network was performed using the “WGCNA” package (version 1.71) in R software (version 4.3.1), with the top 5000 genes exhibiting the highest median absolute deviation. To determine the appropriate soft threshold (β = 5), we utilized the pickSoftThreshold function to compute the network’s topological fit indices. Subsequently, we applied the soft threshold (β = 5) to generate the adjacency matrix, with a resulting R^2^ value of 0.85 achieved by calculating the Pearson’s correlation coefficient between any two genes within the matrix. We performed gene module analysis using the blockwiseModules function. To identify modules (minimum size = 30), hierarchical clustering and dynamic tree-cutting functions were employed. Each co-expression module was assigned a name corresponding to its color. By using the corPvalueStudent function and the cor function to assess the *p*-values and Pearson’s correlation coefficients of the module eigengenes (MEs) in relation to the disease traits, the key modules most relevant to HCM were identified.

### 4.4. Identification of Immune-Related RNAs and ceRNA Network Construction

The AnnoProbe package was used to annotate the genes from key modules, because its annoGene function can distinguish between mRNAs and lncRNAs. We downloaded a list of immune-related genes from the gene list resources in the Immunology Database and Analysis Portal (ImmPort) (https://www.immport.org/ (accessed on 1 June 2022)). Then, we identified the immune-related mRNAs and the immune-related lncRNAs by intersecting the list from ImmPort and the genes from key modules.

The construction of the ceRNA network involved the utilization of immune-related lncRNAs and mRNAs. To obtain interaction data between miRNAs, mRNAs, and lncRNAs, we employed the miRcode database, which provided both predicted and experimentally validated information. Competing lncRNA–mRNA pairs were identified by two criteria: the existence of regulatory miRNAs shared between the lncRNA and mRNA, and a positive correlation in the expression levels of the mRNA and lncRNA (Pearson’s correlation > 0.75). The ceRNA network derived from these identified lncRNA–mRNA pairs was visualized using Cytoscape 3.9.1 software.

### 4.5. Cell Culture and HCM Cell Model

AC16 cells were purchased from SUNNCELL. The cells were cultured in Dulbecco’s modified Eagle’s medium (DMEM; GENOM, Hangzhou, China) supplemented with 10% fetal bovine serum (FBS; ExCell, Suzhou, China), cultured for 24 h at 37 °C in a humidified incubator under 5% CO_2_. ISO group cells were incubated with medium containing 10 µM ISO [39] (MilliporeSigma, Burlington, MA, USA) for 24 h, and cells treated with PBS were set as the control group. Two *MIR210HG* siRNAs were fabricated by Genechem, and si-NC-FAM was transfected as a negative control. For transient transfection, a mixture of siRNA (60 nM), Opti-MEM medium (ThermoFisher Scientific, Waltham, MA, USA), and GP-transfect-Mate (Genechem, Shanghai, China) was dispensed into each well of 24-well cell culture dishes containing AC16 cells.

### 4.6. Measurement of Cell Surface Area

To determine changes in cell size, cells were treated with 4% paraformaldehyde at room temperature (RT) for 20 min, 0.1% Triton X-100 PBS was used for permeabilization, and then actin tracker (1:100, Beyotime, Shanghai, China) was added to the cells and incubated at RT for 1 h. Finally, Hoechst 33342 (1:100, Beyotime, Shanghai, China) was applied to the cells at RT for 5 min. Cells were randomly selected from different groups. Three fields per dish image were captured and imaged using a fluorescence microscope (Keyence, BZ-X800E) at a magnification of ×40. The cell surface area was measured using ImageJ 1.46r.

### 4.7. Quantitative Real-Time PCR (qRT-PCR)

Total RNA was extracted as previously described [40]. Briefly, the total RNA of the fresh AC16 cells was extracted using TRIzol reagent (Tsingke, Beijing, China). To detect the expression of lncRNAs and mRNAs, the HiScript III RT SuperMix for qPCR Kit (Vazyme, Nanjing, China) was used to synthesize the cDNA with 1 µg of total RNA. Then, 6.5 µL of 50-fold-diluted cDNA, 7.5 µL of qPCR SYBR Master Mix (Vazyme, Nanjing, China) and 1 µL of 10 mM primers were mixed in a 96-well plate. The method of thermal profiling followed the kit instructions, and 18S was used as an internal reference. To detect the miRNAs’ expression levels, reverse transcription was performed using the miRcute Plus miRNA First-Strand cDNA Kit (Tiangen Biotech Co.m Ltd., Beijing, China), according to the manufacturer’s instructions. U6 was used as an internal reference. For each PCR reaction, Dissociation Curve 1.0 software (Applied Biosystems, Waltham, MA, USA) was used to analyze the dissociation curves to detect and eliminate possible primer dimers and nonspecific amplification. The 2^−∆∆Ct^ method was used to calculate the relative abundance of lncRNAs, mRNAs, and miRNAs, accounting for gene-specific efficiencies and normalized to the mean expression of the abovementioned index [41]. The primers used in this study are listed in Appendix A. 

### 4.8. CIBERSORTx and Statistical Analysis

To investigate the disparities in immune cell composition between HCM and healthy controls, we employed CIBERSORTx (https://cibersortx.stanford.edu/ (accessed on 16 May 2022)) to examine whole-genome expression profiles. Utilizing the LM22 signature as a reference and conducting 1000 permutations, we determined the proportion of each immune-cell type in the samples from both groups. We assessed the correlation between lncRNAs/mRNAs and the proportion of immune cells using Pearson’s correlation coefficients, considering a *p*-value < 0.05 as statistically significant. The Pearson’s correlation coefficients were calculated using the cor function, and the *p*-values were calculated using the rcorr function.

## 5. Conclusions

In this study, RNA-Seq datasets of heart tissues from HCM patients were investigated to identify immune-related lncRNA–mRNA co-expression pairs. Through thorough bioinformatics screening and analysis, we identified an immune-related lncRNA–mRNA pair (*MIR210HG–BPIFC*) that was consistently decreased in HCM, as confirmed by the analysis of the validation dataset and experiments in an HCM cell model. Further immune cell infiltration analysis suggested that the *MIR210HG–BPIFC* pair is potentially involved in the infiltration of naïve CD4^+^ T cells and CD8^+^ T cells during the progression of HCM. These findings not only identify decreased expression of the *MIR210HG–BPIFC* pair as a novel biomarker of HCM, but also suggest that this immune-related lncRNA–mRNA co-expression pair could be a new therapeutic target for the treatment of HCM. 

## Figures and Tables

**Figure 1 ijms-25-02816-f001:**
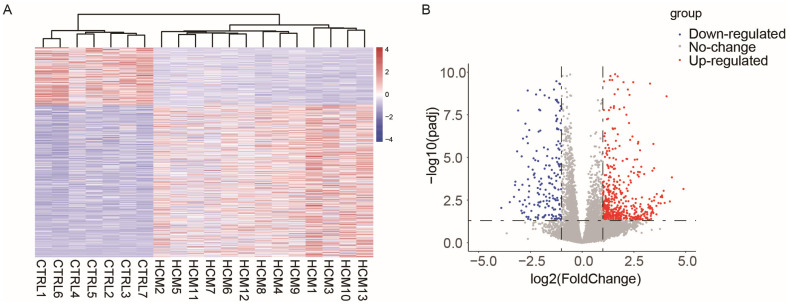
Heatmap and volcano plot illustrating the DEGs identified in the GSE180313 dataset: (**A**) Heatmap showing all DEGs. (**B**) Volcano plot. HCM: hypertrophic cardiomyopathy; CTRL: disease-free heart; DEGs: differentially expressed genes.

**Figure 2 ijms-25-02816-f002:**
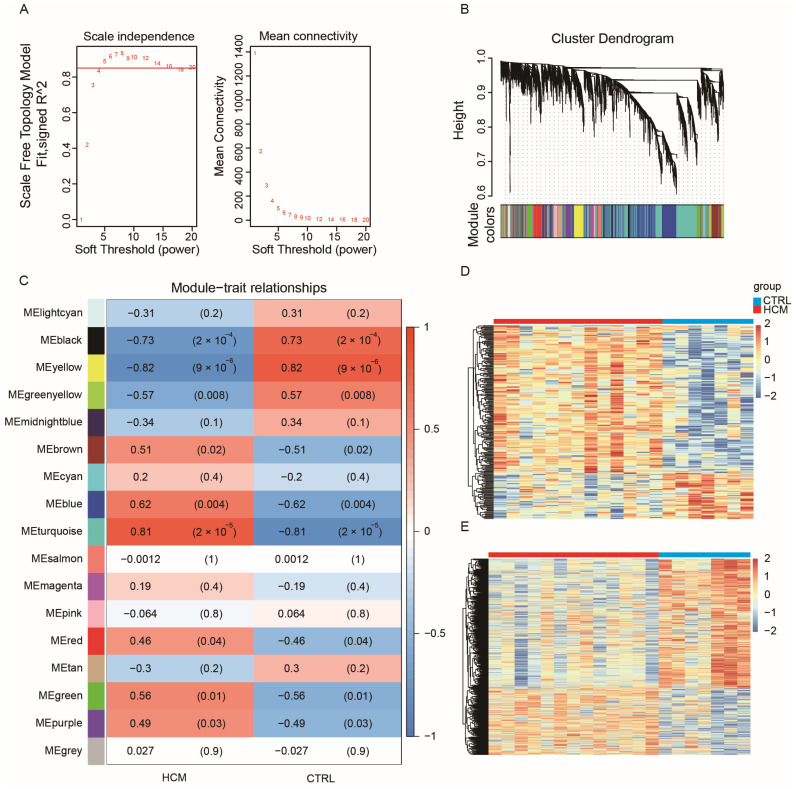
WGCNA for GSE180313: (**A**) Analysis of the scale-free index and mean connectivity was carried out for threshold powers ranging from 1 to 20. We chose a β value of 5 as the soft threshold; this choice led to a scale-free R^2^ value of 0.85. (**B**) Cluster dendrogram of genes in the WGCNA network, with assigned module colors. (**C**) Analysis of module–clinical trait associations. Each column represents a clinical trait, while each row represents a module. Each grid cell contains the correlation (left) and *p*-value (right). The table is color-coded based on the correlation, as shown in the color legend. The correlation coefficients range from −1 to 1, and the association is positively related to the absolute value. (**D**) Heatmap of gene expression profiles in the yellow module. (**E**) Heatmap of gene expression profiles in the turquoise module.

**Figure 3 ijms-25-02816-f003:**
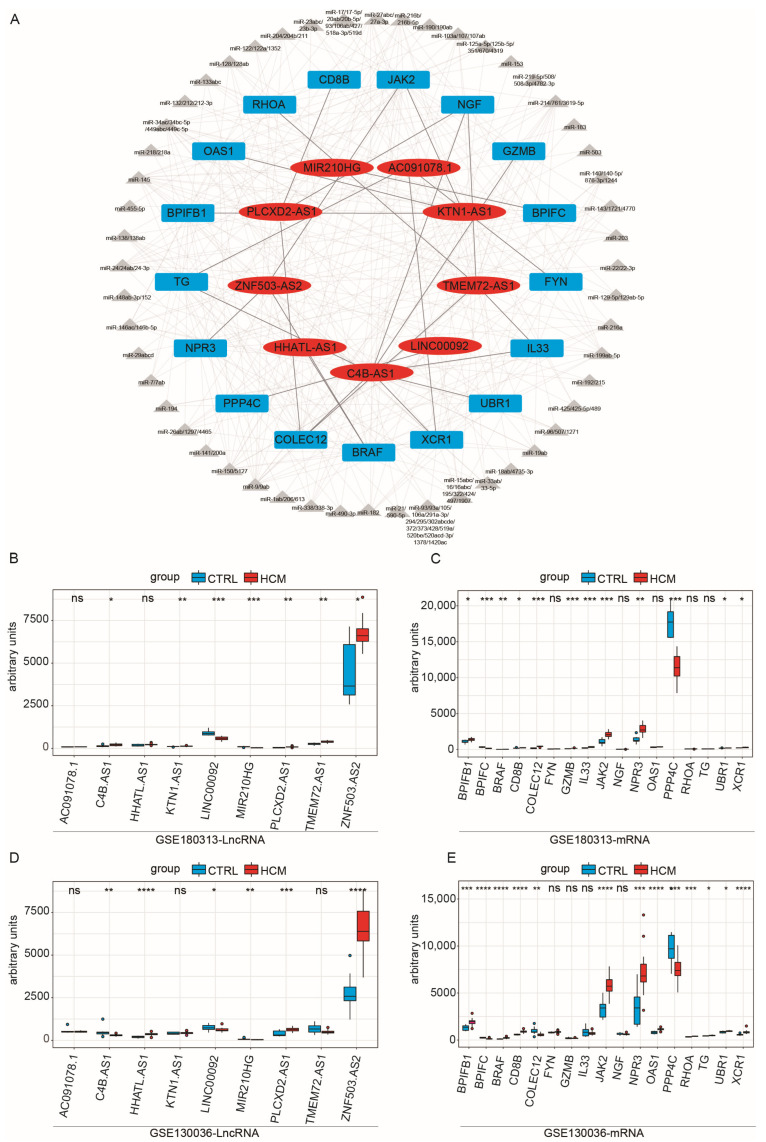
Correlation test of lncRNA–mRNA pairs with disease characteristics in the GSE180313 and GSE130036 datasets: (**A**) The network of immune-related ceRNAs. The red ovals are lncRNAs, the gray triangles are miRNAs, and the blue cubes are mRNAs. (**B**) Expression levels of lncRNAs included in the ceRNA network among the HCM and CTRL samples in the GSE180313 training dataset. (**C**) Expression levels of mRNAs included in the ceRNA network among the HCM and CTRL samples in the GSE180313 training dataset. (**D**) The lncRNAs included in the ceRNA network among the HCM and CTRL samples’ expression levels in the GSE130036 validation dataset. (**E**) The mRNAs included in the ceRNA network among the HCM and CTRL samples’ expression levels in the GSE130036 validation dataset. * *p*-Value < 0.05, ** *p*-value < 0.01, *** *p*-value < 0.001, **** *p*-value < 0.0001; ns: no significance.

**Figure 4 ijms-25-02816-f004:**
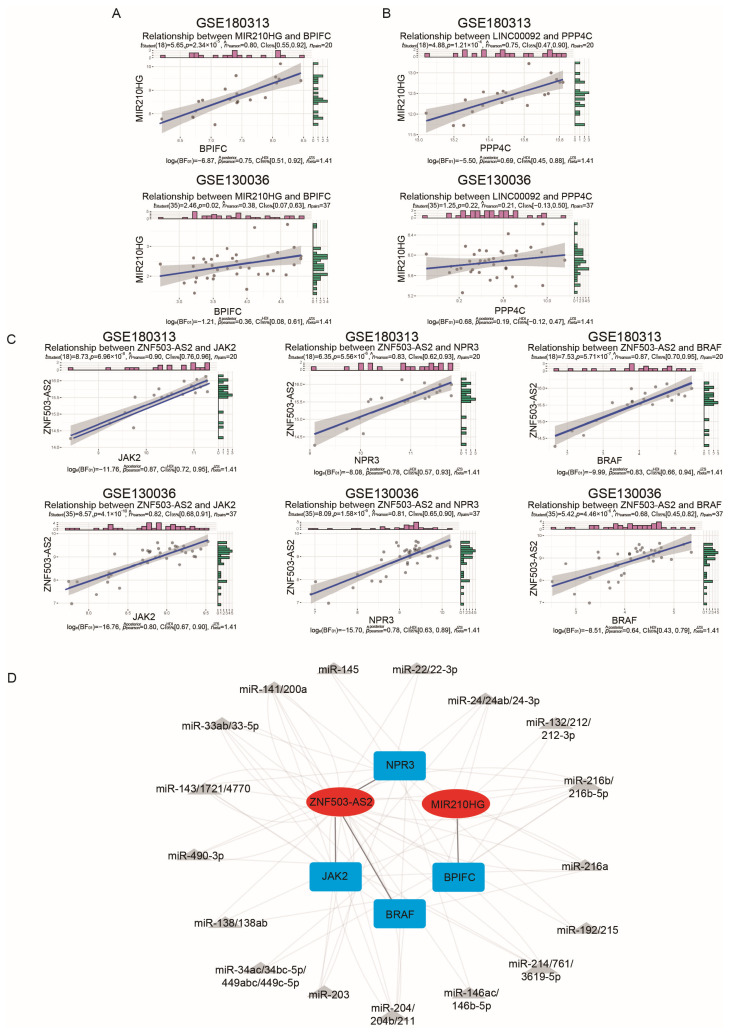
Validation of the immune-related ceRNA network: (**A**–**C**) Validation of correlations of immune-related lncRNA–mRNA pairs in the GSE180313 training dataset and the GSE130036 validation dataset. The pink and green bars showed the expression levels of the indicated genes and the blue lines indicated the linear regression line. (**D**) The ceRNA network confirmed by the GSE180313 training dataset and the GSE130036 validation dataset. The red ovals are lncRNAs, the gray triangles are miRNAs, and the blue cubes are mRNAs.

**Figure 5 ijms-25-02816-f005:**
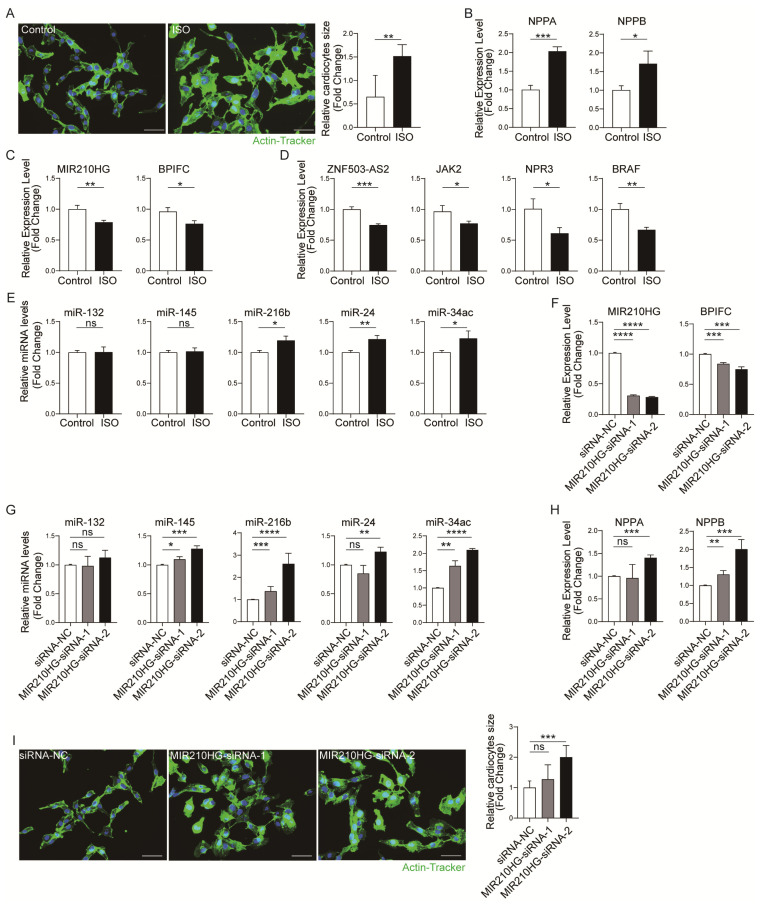
Validation of the immune-related lncRNA–mRNA pairs in vitro: (**A**) Actin-tracker staining results of the control and ISO groups. (**B**) The mRNA expression levels of *Nppa* and *Nppb* in the control and ISO groups. (**C**,**D**) The expression levels of lncRNA–mRNA pairs in the control and ISO groups. (**E**) The expression levels of *MIR210HG*-regulated immune-related miRNAs in the control and ISO groups. (**F**) The expression levels of the *MIR210HG–BPIFC* pair in the siRNA-NC, siRNA-1, and siRNA-2 groups. (**G**) The expression levels of MIR210HG-regulated immune-related miRNA in the siRNA-NC, siRNA-1, and siRNA-2 groups. (**H**) The mRNA expression levels of *Nppa* and *Nppb* in the siRNA-NC, siRNA-1, and siRNA-2 groups. (**I**) Actin-tracker staining results of the siRNA-NC, siRNA-1, and siRNA-2 groups. * *p*-Value < 0.05, ** *p*-value < 0.01, *** *p*-value < 0.001, **** *p*-value < 0.0001, ns: no significance. Control: normal cultured AC16 cells; ISO: ISO-induced AC16 cells; siRNA-NC: NC-FAM-induced AC16 cells; MIR210HG-siRNA-1: one of the siRNAs targeting *MIR210HG*-induced AC16 cells; MIR210HG-siRNA-2: the other siRNA targeting *MIR210HG*-induced AC16 cells. Scale bar = 50 µm.

**Figure 6 ijms-25-02816-f006:**
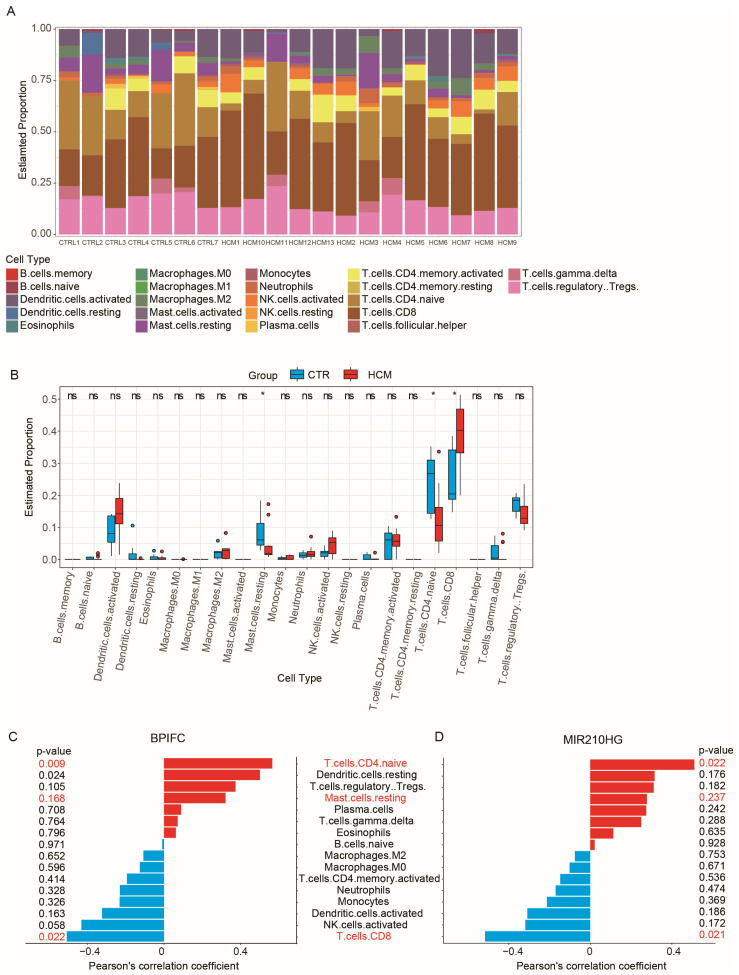
Analysis of immune cell infiltration: (**A**) The 22 immune cells’ proportions in HCM and CTRL samples. (**B**) The immune cells’ differential expression analysis in the CTRL and HCM groups. (**C**) Pearson’s correlation analysis of *BPIFC* and the infiltrating immune cells; statistically significant immune cells are marked in red, with *p*-values < 0.05. (**D**) Pearson’s correlation analysis of *MIR210HG* and the infiltrating immune cells; statistically significant immune cells are marked in red, with *p*-values < 0.05. * *p*-Value < 0.05, ns: no significance.

**Figure 7 ijms-25-02816-f007:**
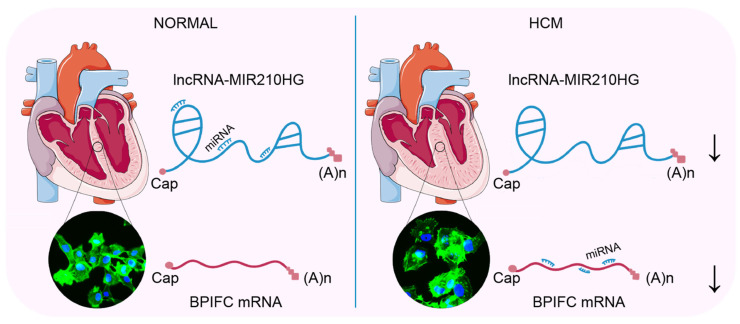
The lncRNA *MIR210HG* regulates the mRNA *BPIFC* via the miRNAs *miR-216b*, *miR-24*, *miR-34c*, etc. Comparing with the normal heart, the decreased expression of lncRNA *MIR210HG* prompts more binding of the mediating miRNAs (*miR-145*, *miR-216b*, *miR-24*, *miR-34c*, etc) with the mRNA of *BPIFC* and induces the degradation of *BPIFC*, associated with the hypertrophy of cardiomyocytes in HCM. The arrows indicated the decreased expression.

**Figure 8 ijms-25-02816-f008:**
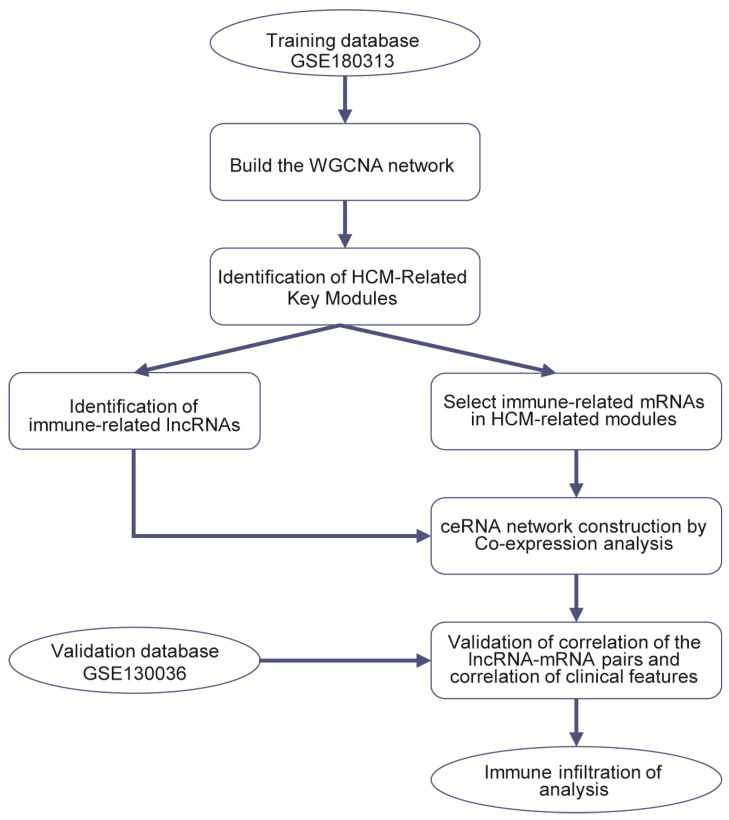
Study flowchart.

**Table 1 ijms-25-02816-t001:** The lncRNA–mRNA pairs with |correlation coefficient| > 0.75 in the co-expression analysis.

Immune-Related mRNA	Immune-Related lncRNA	Correlation Coefficient
OAS1	KTN1-AS1	0.789029469
JAK2	KTN1-AS1	0.752564718
BPIFB1	KTN1-AS1	0.858522985
RHOA	TMEM72-AS1	0.758943191
NGF	TMEM72-AS1	0.773519315
IL33	TMEM72-AS1	0.783806615
COLEC12	TMEM72-AS1	0.761881119
PPP4C	LINC00092	0.75444876
TG	C4B-AS1	0.835504137
GZMB	C4B-AS1	0.791438674
NGF	C4B-AS1	0.763980651
IL33	C4B-AS1	0.786158581
COLEC12	C4B-AS1	0.792932936
UBR1	C4B-AS1	0.798578035
XCR1	C4B-AS1	0.767467957
BRAF	HHATL-AS1	0.855393248
JAK2	ZNF503-AS2	0.899336755
NPR3	ZNF503-AS2	0.831443126
BRAF	ZNF503-AS2	0.871300503
NGF	PLCXD2-AS1	0.77646062
COLEC12	PLCXD2-AS1	0.799703687
CD8B	PLCXD2-AS1	0.761688487
BPIFC	MIR210HG	0.799484632
FYN	AC091078.1	0.780037819
TG	AC091078.1	0.833008855
XCR1	AC091078.1	0.766529515

## Data Availability

All figures and data used to support this study are included in this article.

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
