# Peer review of "Transcriptomic Analyses and Experimental Validation Identified Immune-Related lncRNA–mRNA Pair MIR210HGBPIFC Regulating the Progression of Hypertrophic Cardiomyopathy"

_ijms, 2024, doi:10.3390/ijms25052816_

Round 1

Reviewer 1 Report (New Reviewer)

Comments and Suggestions for Authors

Authors have written a good manuscript on identification of key lncrna-mrna involved in hypertrophic cardiomyopathy using bioinformatics analysis. The study then validated the role of MIR210HG-BPIFC using AC16 cells which confirms that the pair potentially involved in the infiltration of naïve CD4+ T cells and CD8+ T cells. However, there are few issues needs to be resolved by the authors before acceptance:

1. Figure 5 is quite confusing especially finding the figure was poorly explained and labelled. eg; in G for example did all the different types of siRNA reflected by the gene name written above the bar? for Figure 5(I), what gene actually was silenced? since it was just labelled as  "siRNA1 and 2". This figure has to be improved>

2. What was the relevant of measurement of cell surface area in relation with hypertrophic cardiomyopathy?

3. The current conclusion seems weak. Conclusion can be written at such did the research meet the study objectives and what is relevance of the study findings towards community.

Author Response

Reviewer #1: Authors have written a good manuscript on identification of key lncrna-mrna involved in hypertrophic cardiomyopathy using bioinformatics analysis. The study then validated the role of MIR210HG-BPIFC using AC16 cells which confirms that the pair potentially involved in the infiltration of naïve CD4+ T cells and CD8+ T cells. However, there are few issues needs to be resolved by the authors before acceptance: 1. Figure 5 is quite confusing especially finding the figure was poorly explained and labelled. eg; in G for example did all the different types of siRNA reflected by the gene name written above the bar? for Figure 5(I), what gene actually was silenced? since it was just labelled as "siRNA1 and 2". This figure has to be improved> RE: Following the reviewer’s suggestion, we added the silenced gene name in Figure 5, and figure legend in line 261-262. Furthermore, we added the additional instructions of two siRNAs in Result in line 202, and Materials and Methods in line 447. The revised Figure 5 is shown as below. 2. What was the relevant of measurement of cell surface area in relation with hypertrophic cardiomyopathy? RE: Cardiomyocyte hypertrophy is an early response of the heart to stress. In animal models, an increase in cardiomyocyte volume is an early marker of remodeling, occurring in response to mechanical stretch during the transition to heart failure. At a macroscopic level, increased LV thickness is a manifestation of cardiomyocyte hypertrophy and expansion of the extracellular matrix, both of which play an integral role in the transition from compensated cardiac hypertrophy to clinical HF. Many studies of hypertrophic cardiomyopathy have used cell area as a marker for cardiomyocyte hypertrophy in vitro, such as “Coelho-Filho, et al. Circulation, 2013 (PMID: 23912910)” [1] and “Czibik, et al. Circulation, 2021 (PMID: 34162223)” [2]. [1] Coelho-Filho OR, Shah RV, Mitchell R, et al. Quantification of cardiomyocyte hypertrophy by cardiac magnetic resonance: implications for early cardiac remodeling. Circulation. 2013 Sep 10;128(11):1225-33. DOI: 10.1161/CIRCULATIONAHA.112.000438. PMID: 23912910. [2] Czibik G, Mezdari Z, Murat Altintas D, et al. Dysregulated Phenylalanine Catabolism Plays a Key Role in the Trajectory of Cardiac Aging. Circulation. 2021 Aug 17;144(7):559-574. DOI: 10.1161/CIRCULATIONAHA.121.054204. PMID: 34162223. 3. The current conclusion seems weak. Conclusion can be written at such did the research meet the study objectives and what is relevance of the study findings towards community. RE: We have followed the reviewer’s suggestion and rewrote the Conclusion as follows (line 488-500): In this study, RNA-seq datasets of heart tissues from HCM patients were investigated to found immune-related lncRNA-mRNA co-expression pairs. By thoroughly bioinformatic screening and analysis, we identified an immune-related lncRNA–mRNA pair (MIR210HG-BPIFC) correlatedly decreased in HCM, which confirmed by the analysis of validating dataset and experiments in HCM cell model. Further immune cell infiltration analysis suggested MIR210HG-BPIFC pair is potentially involved in the infiltration of naïve CD4+ T cells and CD8+ T cells during the progression of HCM. These findings not only identified decreased expression of MIR210HG-BPIFC pair as a novel biomarker for HCM, but also suggested this immune-related lncRNA-mRNA co-expression pair could be a new therapeutic target for the treatment of HCM.

Reviewer 2 Report (New Reviewer)

Comments and Suggestions for Authors

I have carefully reviewed the manuscript entitled "Transcriptomic Analyses and Experimental Validation Identified Immune-Related lncRNA-mRNA Pair MIR210HG-BPIFC Regulating the Progression of Hypertrophic Cardiomyopathy" which employs a bioinformatics approach to identify potential lncRNA/mRNA pairs implicated in the pathogenesis of hypertrophic cardiomyopathy (HCM), followed by experimental validation in the AC16 human cardiomyocyte cell line treated with isoprenaline (ISO). The manuscript is logically structured, and the experimental methodology is robust, offering valuable insights into the molecular mechanisms of HCM.

Major Concern:

1.     The manuscript extensively utilizes bioinformatic analyses to explore the interactions between lncRNAs and mRNAs in HCM. However, it solely relies on experiments conducted using AC16 cells, omitting analyses on CD4 and CD8 T cells. This omission raises questions about the connection between the MIR210HG/BPIFC pair and T cell infiltration in HCM. This needs to be discussed.

2.     Given that the study aimed to identify immune-related lncRNA/mRNA pairs, the impact of the MIR210HG/BPIFC pair on immune phenomena within HCM remains ambiguous. This needs to be discussed.

Minor Concerns:

1.     In the introduction, there is an inaccurately presented statement: " IL-6 which is a typical cytokine associated with inflammation can also in-56 duce the differentiation of naïve T cells into CD8+ T cells [8]." The reference provided does not support this claim, necessitating correction or further clarification to ensure accuracy.

2.     Section 2.4 contains a grammatical error in the description of qRT-PCR results: "qRT-PCR result 202 shown that the mRNA level of BPIFC was decreased correlated with knockdown of 203 MIR210HG (Figure 5F).".

3.     In section 4.5, the presentation of information regarding ISO appears to be misleading: " AC16 cells are purchased from SUNNCELL, and induced AC16 cells (ISO) were 431 cultured in Dulbecco’s modified Eagle’s medium (DMEM; GENOM, China).”

I recommend addressing these concerns to enhance the manuscript's contribution to our understanding of HCM. The study's innovative approach and findings are commendable, yet refining these areas will undoubtedly strengthen the paper's impact and scientific rigor.

Author Response

Major Concern:

  1. The manuscript extensively utilizes bioinformatic analyses to explore the interactions between lncRNAs and mRNAs in HCM. However, it solely relies on experiments conducted using AC16 cells, omitting analyses on CD4 and CD8 T cells. This omission raises questions about the connection between the MIR210HG/BPIFC pair and T cell infiltration in HCM. This needs to be discussed.

RE: We thank Reviewer #2’s careful reading of our manuscript and appreciate his/her highly constructive comments. By deeply analysis, our work identified the connection between the MIR210HG/BPIFC pair and T cell infiltration of immune response in HCM.

First, T-cell infiltration has been found in the published HCM heart single-cell data (Chaffin, et al. Nature, 2022 (PMID: 35732739)). In the cluster of T-cells, the proportion of the subgroups with increased CD8+ marker genes expression was increased, indicated that the infiltration of CD8+ T cell was increased in HCM, which was consistent with our finding. Second, since the MIR210HG-BPIFC pair was screened through a list of immune-related genes from the gene list resources in the Immunology Database and Analysis Portal (ImmPort) (https://www.immport.org/ (accessed on 1 June 2022)) (line 420-424), we defined the MIR210HG-BPIFC pair as an “immune-related” pair. Particularly, MIR210HG has been reported associated with immune response (Yadav, et al. Life Sci, 2024 (PMID: 38242493.)). Finally, Pearson’s correlation analysis of MIR210HG and BPIFC with the infiltrating immune cells results (Figure 6C-D) shown the connection between MIR210HG and BPIFC with naïve CD4+ T cells and CD8+ T cells. We have added the above content in the Discussion part of the revised manuscript (line 368-371).

  1. Given that the study aimed to identify immune-related lncRNA/mRNA pairs, the impact of the MIR210HG/BPIFC pair on immune phenomena within HCM remains ambiguous. This needs to be discussed.

RE: Regarding the reviewer’s concern on the impact of the MIR210HG/BPIFC pair on immune phenomena within HCM, we have updated the Discussion focusing on the mechanism of how the pair regulates the immune response in HCM. The expression changes of MIR210HG/BPIFC pair were correlated with the immune-related miRNAs in AC16 cardiomyocyte hypertrophy model (Figure 5E and Figure S2A). Based on our bioinformatic analysis and experimental data, we speculated that the decreased MIR210HG/BPIFC pair in AC16 may regulate the innate immune response cytokines and affect the recruitment of naïve CD4+ T cells and CD8+ T cells. However, the detailed molecular mechanism is still obscure and needs further studies. We have added the above content in the Discussion part of the revised manuscript (line 371-375 and 381).

Minor Concerns:

  1. In the introduction, there is an inaccurately presented statement: " IL-6 which is a typical cytokine associated with inflammation can also in-56 duce the differentiation of naïve T cells into CD8+ T cells [8]." The reference provided does not support this claim, necessitating correction or further clarification to ensure accuracy.

RE: Thank you for correcting our mistakes. We changed the reference 8 into reference 9 “Yang R, Masters AR, Fortner KA, et al. IL-6 promotes the differentiation of a subset of naive CD8+ T cells into IL-21-producing B helper CD8+ T cells. J Exp Med 2016, 213(11),2281-2291.” in line 57. Moreover, we added the reference 8 “Kumar S, Wang G, Zheng N, et al. HIMF (Hypoxia-Induced Mitogenic Factor)-IL (Interleukin)-6 Signaling Mediates Cardiomyocyte-Fibroblast Crosstalk to Promote Cardiac Hypertrophy and Fibrosis. Hypertension 2019, 73(5),1058-1070.” in line 56.

  1. Section 2.4 contains a grammatical error in the description of qRT-PCR results: "qRT-PCR result 202 shown that the mRNA level of BPIFC was decreased correlated with knockdown of 203 MIR210HG (Figure 5F).".

RE: Thank you for correcting our mistakes. We changed “decreased correlated” into “correlatedly decreased” in line 203-204.

  1. In section 4.5, the presentation of information regarding ISO appears to be misleading: " AC16 cells are purchased from SUNNCELL, and induced AC16 cells (ISO) were 431 cultured in Dulbecco’s modified Eagle’s medium (DMEM; GENOM, China).”

RE: Following the reviewer’s suggestion, we changed “AC16 cells are purchased from SUNNCELL, and induced AC16 cells (ISO) were cultured in Dulbecco’s modified Eagle’s medium (DMEM; GENOM, China).” into “AC16 cells are purchased from SUNNCELL. The cells were cultured in Dulbecco’s modified Eagle’s medium (DMEM; GENOM, China) supplemented with 10% fetal bovine serum (FBS; ExCell, China) at 37℃ in a humidified incubator under 5% CO2. ISO group cells are incubated with medium containing 10 μM ISO [38] (Sigma, USA) for 24 h, and cells treated with PBS are set as Control group.” in line 443-447.

This manuscript is a resubmission of an earlier submission. The following is a list of the peer review reports and author responses from that submission.

Round 1

Reviewer 1 Report

Comments and Suggestions for Authors

The study entitled ‘Transcriptomic Analyses Identified lncRNA–mRNA Pair MIR210HG-BPIFC regulating the progression of Hypertrophic Cardiomyopathy through immune cell infiltration’ by Juan Zhang and coworkers presents a new lncRNA-mRNA pair potentially important for occurrence of hyperthophic cardiomyopathy (HCM). The Authors analysed publically available RNA-seq data from healthy tissues and HCM patients’ samples, to find immune- related lncRNAs co-expressed with mRNAs in HCM.

Additionally, microRNAs that are common for the identified lncRNA-mRNA pairs were selected. Few lncRNA-mRNA pairs were verified using additional publically available RNA-seq dataset, as well as by QPCR performed on samples from ischemic cells model .Finally, pair MIR210HG-BPIFC was selected as significantly important, and immune infiltration analyses were performed to prove the importance of their regulation of  immune cells composition.

This work is mainly based on an integrated  bioinformatics approach undertaken to analyse publically available RNA-seq data from HCM patients, showing lncRNA-mRNA pair potentially important for occurrence of HCM , yet in my opinion the functional relevance of this finding is missing.  The conclusions are based mainly on correlation between expression of lncRNA-mRNA pairs and their association in  13 HCM samples . And as the Authors have noticed  MIR210HG seems to play multiple roles in various diseases (lines 282-285) acting with different of microRNAs - the experimental data to verify its specific involvement in HCM is needed and would be beneficial to improve quality of the paper.

Also, the gene expression data showing immune cell distribution in ISO model analysed by CIBERSORTx expression data are too preliminary to verify if MIR210HG-BPIFC are indeed  “regulating the progression of Hypertrophic Cardiomyopathy through immune cell infiltration”, as stated in the title. These results are not described and discussed well.

The QPCR analysis of expression of lncRNA-mRNA pairs performed to validate the identified ceRNA network is not described well in the Materials and Methods section. The reference genes are not included, as well as methods of calculation of the relative gene expression, and the supportive references are missing.

I have also found the quality of the Figure 3, and 5 to low to read genes and cell types names and values. Similarly, regarding Figure 4B the charts are to small to read the results. Therefore, figures and charts should be replaced with high resolution images, where necessary.

Minor issues:

Figures 1 to 6 have two descriptions, on top and bottom of each  figure.

Line 85: result –s is missing

Line 88 we- capital letter

In the volcano plot islet “no-change” I suggest to change to e.g. “unchanged”

Line 173-176 “... AC16 cells, which were derived from the fusion of primary cells from adult  ventricular heart tissue with SV40-transformed uridine trophotrophic human fibroblasts, and can be used to study cardiac gene expression and function at cellular, organ, and molecular levels during normal development and under pathological conditions”- please provide references.

Line 177 and 354- 10uL change to µl

Line 189-190 Asterisks missing in description of significantly important values

Line 192 “Validation of the immune related ceRNA network”  I suggest to add “in HCM”

Line 194 a dot is missing before (B)

Line 200-more references is needed if the method is “commonly used”

Line 271, and 27 “four candidate lncRNAs”-this is confusing because  in the text the Authors mentioned about two lncRNAs

Line 287 pairs- should be pair

Line 257-258 This sentence seems to be isolated from the previous ones.

Line 325-3 dot missing at the end of the sentence

Comments on the Quality of English Language

Minor editing of English language required.

Author Response

Dear Reviewer:

We greatly appreciate your valuable suggestions in improving this paper. Based on the suggestion, we have completed the revision of the incorrect descriptions. In addition, we have carefully considered the comments outlined in the reviews and revised our manuscript. We changed the title of the revised manuscript into “Transcriptomic and Experimental Analyses Identified lncRNA–mRNA Pair MIR210HG-BPIFC as an Immune-Related Biomarker for Hypertrophic Cardiomyopathy” in order to better explain the main idea of our article. Here is the list of responses to the comments containing a detailed point-by-point description:

Comments:

The study entitled ‘Transcriptomic Analyses Identified lncRNA–mRNA Pair MIR210HG-BPIFC regulating the progression of Hypertrophic Cardiomyopathy through immune cell infiltration’ by Juan Zhang and coworkers presents a new lncRNA-mRNA pair potentially important for occurrence of hyperthophic cardiomyopathy (HCM). The Authors analysed publically available RNA-seq data from healthy tissues and HCM patients’ samples, to find immune- related lncRNAs co-expressed with mRNAs in HCM.

Additionally, microRNAs that are common for the identified lncRNA-mRNA pairs were selected. Few lncRNA-mRNA pairs were verified using additional publically available RNA-seq dataset, as well as by QPCR performed on samples from ischemic cells model. Finally, pair MIR210HG-BPIFC was selected as significantly important, and immune infiltration analyses were performed to prove the importance of their regulation of immune cells composition.

This work is mainly based on an integrated bioinformatics approach undertaken to analyse publically available RNA-seq data from HCM patients, showing lncRNA-mRNA pair potentially important for occurrence of HCM, yet in my opinion the functional relevance of this finding is missing.  The conclusions are based mainly on correlation between expression of lncRNA-mRNA pairs and their association in 13 HCM samples. And as the Authors have noticed MIR210HG seems to play multiple roles in various diseases (lines 282-285) acting with different of microRNAs - the experimental data to verify its specific involvement in HCM is needed and would be beneficial to improve quality of the paper. Also, the gene expression data showing immune cell distribution in ISO model analysed by CIBERSORTx expression data are too preliminary to verify if MIR210HG-BPIFC are indeed “regulating the progression of Hypertrophic Cardiomyopathy through immune cell infiltration”, as stated in the title. These results are not described and discussed well.

RE:

Since MIR210HG play multiple roles in various diseases acting with different of microRNAs has been confirmed. We further verified the microRNAs related with MIR210HG-BPIFC pair.

Total RNA was isolated from AC16 cells using TRIZol reagent according to the manufacturer’s instructions (Tsingke, Beijing, China). Reverse transcription was performed using the miRcute Plus miRNA First-Strand cDNA Kit (Tiangen Biotech Co. Ltd, Beijing), according to the manufacturer’s instructions. The levels of microRNAs were detected by QRT-PCR using miRcute Plus miRNA qPCR Kit (Tiangen Biotech Co. Ltd, Beijing). U6 was used as an internal reference. For each PCR reaction, Dissociation Curve 1.0 Software (Applied Biosystems) was used to analyze dissociation curves in order to detect and eliminate possible primer-dimers and nonspecific amplification. The 2-∆∆Ct method was used to calculate the relative abundance of miRNA, accounting for gene-specific efficiencies and normalized to the mean expression of the above-mentioned index. The primers used in the experiment are listed as follow:

Table 1 Primers used in current experiment

Gene

Primer sequence

miR-145

Forward: CGTCCAGTTTTCCCAGGAATCCCT

miR-216b

Forward: CGCTAATCTCTGCAGGCAACTGT

miR-216b-5p

Forward: GCCAAATCTCTGCAGGCAAATGTGA

miR-24

Forward: CTGGCTCAGTTCAGCAGGAACAG

miR-24ab

Forward: CCGGTGCCTACTGAGCTGATAACA

miR-24-3p

Forward: CGGTGCCTACTGAGCTGATATCAGT

mir-34ac

Forward: CTGGCAGTGTCTTAGCTGGTTGT

miR-34bc-5p

Forward: GCCGTAGGCAGTGTCATTAGCTGA

miR-449abc

Forward: CGTGGCAGTGTATTGTTAGCTGGT

miR-449c-5p

Forward: TAGGCAGTGTATTGCTAGCGGCT

miR-122

Forward: CGCCGAACGCCATTATCACACTA

U6

Forward: CACGCAAATTCGTGAAGCGTTCCA

The qPCR result shown in figure 1 indicated that, miRNA-216b, miRNA-216b-5p, miRNA-24, miRNA-24ab, miRNA-34bc-5p, miRNA-449abc, and miRNA-449c-5p was showing the consistent expression trend.

figure 1 Expression of lncRNA–mRNA pairs related miRNA in the ISO induced AC16 cell model. *p-value<0.05 ,** p-value<0.01 ,*** p-value<0.001 ; Control, normal cultured AC16 cell; ISO, ISO induced AC16 cell.

Since, the topic of this study was the lncRNA–mRNA Pair participate in the progression of Hypertrophic Cardiomyopathy through immune cell infiltration, we did not add this result in our manuscripts. In addition, we did not explain the function of lncRNA–mRNA Pair MIR210HG-BPIFC regulating the progression of Hypertrophic Cardiomyopathy through immune cell infiltration, we changed the title of the revised manuscript into “Transcriptomic and Experimental Analyses Identified lncRNA–mRNA Pair MIR210HG-BPIFC as an Immune-Related Biomarker for Hypertrophic Cardiomyopathy” in order to better explain the main idea of our article.

The QPCR analysis of expression of lncRNA-mRNA pairs performed to validate the identified ceRNA network is not described well in the Materials and Methods section. The reference genes are not included, as well as methods of calculation of the relative gene expression, and the supportive references are missing.

RE:

We added the detail information of the method of qPCR in our manuscripts as follow:

Total RNA was extracted as previously described [35]. Briefly, total RNA of the fresh AC16 cells was extracted using TRIZol reagent (Tsingke, Beijing, China). The HiScript III RT SuperMix for qPCR kit (Vazyme, Nanjing, China) was performed to synthesize the cDNA with 1 µg of total RNA. Then, 6.5 µl 50-fold dilution cDNA, 7.5 µl qPCR SYBR Master Mix (Vazyme, Nanjing, China) and 1 µl 10 mM primers were mixed in 96-well plate. The method of thermal profile was followed the kit instructions. 18S was used as an internal reference. For each PCR reaction, Dissociation Curve 1.0 Software (Applied Biosystems) was used to analyze dissociation curves in order to detect and eliminate possible primer-dimers and nonspecific amplification. The 2-∆∆Ct method was used to calculate the relative abundance of mRNAs, accounting for gene-specific efficiencies and normalized to the mean expression of the above-mentioned index [36]. The primers used in the study are listed in Table S7.

I have also found the quality of the Figure 3, and 5 to low to read genes and cell types names and values. Similarly, regarding Figure 4B the charts are to small to read the results. Therefore, figures and charts should be replaced with high resolution images, where necessary.

RE: We increased the resolution and changed the layout of the images to improve the quality of the figures.

Minor issues:

  • Figures 1 to 6 have two descriptions, on top and bottom of each figure.
  • Line 85: result –s is missing
  • Line 88 we- capital letter
  • In the volcano plot islet “no-change” I suggest to change to e.g. “unchanged”
  • Line 173-176 “... AC16 cells, which were derived from the fusion of primary cells from adult ventricular heart tissue with SV40-transformed uridine trophotrophic human fibroblasts, and can be used to study cardiac gene expression and function at cellular, organ, and molecular levels during normal development and under pathological conditions”- please provide references.
  • Line 177 and 354- 10uL change to µl 360361
  • Line 189-190 Asterisks missing in description of significantly important values
  • Line 192 “Validation of the immune related ceRNA network” I suggest to add “in HCM”
  • Line 194 a dot is missing before (B)
  • Line 200-more references is needed if the method is “commonly used”
  • Line 271, and 27 “four candidate lncRNAs”-this is confusing because in the text the Authors mentioned about two lncRNAs
  • Line 287 pairs- should be pair
  • Line 257-258 This sentence seems to be isolated from the previous ones.
  • Line 325-3 dot missing at the end of the sentence

RE: Thanks for your good suggestions. We agree with your assessment of the analysis.

  • We deleted the descriptions on top of Figures 1 to 6.
  • We added the “s” behind “result” in line 85.
  • We changed the “we” into “We” in line 88.
  • We changed the “no-change” into “unchanged” in the volcano plot islet.
  • We added the reference [16] Novel cell lines derived from adult human ventricular cardiomyocytes in line 176.
  • We changed the “u” into “µ” in line 177, 361, and 367-8.
  • We added the “*” in line 189-190.
  • We added the “in HCM” behind “Validation of the immune related ceRNA network” in line 192.
  • We added a dot before (B) in line 194.
  • We added the reference [17] Profiling Cell Type Abundance and Expression in Bulk Tissues with CIBERSORTx in line 201.
  • We changed the “four candidate lncRNAs” into “two candidate lncRNAs” in line 272 and 275.
  • We changed the “pairs” into “pair” in line 288.
  • We added “were also included in key modules” in line 258-259.
  • We added a dot in line 325.

Reviewer 2 Report

Comments and Suggestions for Authors

Comments to the Authors:

I have read and reviewed your manuscript entitled " Transcriptomic Analyses Identified lncRNA–mRNA Pair MIR210HG-BPIFC regulating the progression of Hypertrophic Cardiomyopathy through immune cell infiltration". In my opinion, the Authors present valuable data, however, the present form of the paper does not fit for publication in the International Journal of Molecular Sciences journal. The list of my comments and suggestions is presented below:

1.      Do you know the patients' age and gender?

2.      Please add more information about statistical analysis you used in your studies, for example the type of statistical tests used to discuss the results.

3.      Why did you use datasets GSE180313 as training dataset and GSE130036 as validation dataset and not the other way around?

4.      Chapter 4.6.  This chapter does not follow the MIQE guidelines for publication of  qPCR results (concentration of cDNA, method thermal profile). Have you performed primers validation? What the type of method was used to calculate gene expression? Did you use references gene?

5.      Some figures are illegible (for example 2c, 2d, 4b)

6.      There is a lot of editorial errors, i.e. gene names with italic/non-italic, etc.

Author Response

Dear Reviewer:

We greatly appreciate your valuable suggestions in improving this paper. Based on the suggestion, we have completed the revision of the incorrect descriptions. In addition, we have carefully considered the comments outlined in the reviews and revised our manuscript. We changed the title of the revised manuscript into “Transcriptomic and Experimental Analyses Identified lncRNA–mRNA Pair MIR210HG-BPIFC as an Immune-Related Biomarker for Hypertrophic Cardiomyopathy” in order to better explain the main idea of our article. Here is the list of responses to the comments containing a detailed point-by-point description:

Comments:

I have read and reviewed your manuscript entitled " Transcriptomic Analyses Identified lncRNA–mRNA Pair MIR210HG-BPIFC regulating the progression of Hypertrophic Cardiomyopathy through immune cell infiltration". In my opinion, the Authors present valuable data, however, the present form of the paper does not fit for publication in the International Journal of Molecular Sciences journal. The list of my comments and suggestions is presented below:

  1. Do you know the patients' age and gender?

RE:

Since the sample data in GSE180313 has been screened and only query unfiltered sample data can be find, we cannot provide detailed information of data of patients' age and gender in GSE180313. In addition, the information of data of patients' age and gender in GSE130036 has been added in TableS6.

  1. Please add more information about statistical analysis you used in your studies, for example the type of statistical tests used to discuss the results.

RE:

We added more information about statistical analysis in our studies as follows:

We added “To filter out genes with a total count less than 1 in the gene expression matrix, and then perform differential analysis using the DESeq function. Finally, we get the DEGs with the cutoff |Log2 fold change|>1 and adjusted p-value<0.05.” in line 326-328.

We added “We performed gene module analysis using the blockwiseModules function.” In line 336-337.

We added “By corPvalueStudent function and cor function” In line 339-340.

We added “The Pearson correlation coefficients are calculated using the cor function, and the p-values are calculated using the rcorr function.” In line 379-380.

  1. Why did you use datasets GSE180313 as training dataset and GSE130036 as validation dataset and not the other way around?

RE:

The datasets GSE180313 and GSE130036 were both generated from human heart samples. However, upon reading the articles that disclose these two datasets, we found that although both datasets claim to have collected hearts from HCM patients, the diagnoses primarily relied on echocardiography. In the case of GSE180313, the samples were subjected to molecular biology experiments and morphological validation, while GSE130036 samples only underwent RNA-seq validation. Therefore, we have more confidence in the samples from HCM patients in GSE180313. Additionally, since the data published in GSE180313 is more recent, we have chosen it as the training set to construct the WGCNA network.

  1. Chapter 4.6. This chapter does not follow the MIQE guidelines for publication of  qPCR results (concentration of cDNA, method thermal profile). Have you performed primers validation? What the type of method was used to calculate gene expression? Did you use references gene?

RE:

We added the detail information of the method of qPCR in our manuscripts as follow:

Total RNA was extracted as previously described [35]. Briefly, total RNA of the fresh AC16 cells was extracted using TRIZol reagent (Tsingke, Beijing, China). The HiScript III RT SuperMix for qPCR kit (Vazyme, Nanjing, China) was performed to synthesize the cDNA with 1 µg of total RNA. Then, 6.5 µl 50-fold dilution cDNA, 7.5 µl qPCR SYBR Master Mix (Vazyme, Nanjing, China) and 1 µl 10 mM primers were mixed in 96-well plate. The method of thermal profile was followed the kit instructions. 18S was used as an internal reference. For each PCR reaction, Dissociation Curve 1.0 Software (Applied Biosystems) was used to analyze dissociation curves in order to detect and eliminate possible primer-dimers and nonspecific amplification. The 2-∆∆Ct method was used to calculate the relative abundance of mRNAs, accounting for gene-specific efficiencies and normalized to the mean expression of the above-mentioned index [36]. The primers used in the study are listed in Table S7.

In addition, the primer of MIR210HG was refers to the reference [29] “MIR210HG promotes breast cancer progression by IGF2BP1 mediated m6A modification”, the primer of ZNF503-AS2 was designed by NCBI primer blast, and the rest of gene primers were designed by Primer Bank, we added the Primer Bank ID in table S7. Furthermore, primers validation was performed through solubility curve.

  1. Some figures are illegible (for example 2c, 2d, 4b)

RE: We increased the resolution and changed the layout of the images to improve the quality of the figures.

  1. There is a lot of editorial errors, i.e. gene names with italic/non-italic, etc.

RE:

Thank you for correcting our mistakes. We have carefully corrected the similar problem in our manuscript.

Reviewer 3 Report

Comments and Suggestions for Authors

The manuscript presented by Zhang et al. explores the lncRNA and mRNA networks functional in cardiomyopathy that are associated with immune cell infiltration and immune response. The study first analyses the RNAseq datasets derived from normal and diseased individual samples and then using bioinformatics algorithm identifies candidate mRNA-lncRNA pairs. They validate a subset of these in independent dataset and validate their dysregulation in disease model by qRT-PCR. The study executed well. However, I have major and minor concerns with the logic of the workflow.

Major

The approach taken by authors for selecting the genes for input to WGCNA is that they take top 5000 mean absolute deviation of (MAD) GSE180313 dataset. The dataset consist of both normal and diseased samples. Given that most differentially expressed genes (DEG) show most MAD, I expect top 5000 genes to consist of DEGs. However, WGCNA does not recommend using differentially expressed genes as input for coexpression network construction since it does not represent a true functional network of co-expressing genes but rather only dysregulated genes which may not be functional in the disease. It is unclear whether authors removed the normal samples from the input. If not what is the rational for using normal samples. Genes that are differential may not be functional in the disease.

Minor

The resolution of figures need to be increased. Especially in figure 4 the text is not legible.

There are a few minor english language errors that need to be corrected by the authors.

Comments on the Quality of English Language

There are a few minor english language errors that need to be corrected by the authors.

Author Response

Dear Reviewer:

We greatly appreciate your valuable suggestions in improving this paper. Based on the suggestion, we have completed the revision of the incorrect descriptions. In addition, we have carefully considered the comments outlined in the reviews and revised our manuscript. We changed the title of the revised manuscript into “Transcriptomic and Experimental Analyses Identified lncRNA–mRNA Pair MIR210HG-BPIFC as an Immune-Related Biomarker for Hypertrophic Cardiomyopathy” in order to better explain the main idea of our article. Here is the list of responses to the comments containing a detailed point-by-point description:

Comments:

The manuscript presented by Zhang et al. explores the lncRNA and mRNA networks functional in cardiomyopathy that are associated with immune cell infiltration and immune response. The study first analyses the RNAseq datasets derived from normal and diseased individual samples and then using bioinformatics algorithm identifies candidate mRNA-lncRNA pairs. They validate a subset of these in independent dataset and validate their dysregulation in disease model by qRT-PCR. The study executed well. However, I have major and minor concerns with the logic of the workflow.

Major

The approach taken by authors for selecting the genes for input to WGCNA is that they take top 5000 mean absolute deviation of (MAD) GSE180313 dataset. The dataset consist of both normal and diseased samples. Given that most differentially expressed genes (DEG) show most MAD, I expect top 5000 genes to consist of DEGs. However, WGCNA does not recommend using differentially expressed genes as input for coexpression network construction since it does not represent a true functional network of co-expressing genes but rather only dysregulated genes which may not be functional in the disease. It is unclear whether authors removed the normal samples from the input. If not what is the rational for using normal samples. Genes that are differential may not be functional in the disease.

RE:

We believe that the data filtering is necessary, but there is no consensus on which method is most suitable for gene screening. MAD is one of the data filtering methods recommended in the WGCNA official website FAQ (https://horvath.genetics.ucla.edu/html/CoexpressionNetwork/Rpackages/WGCNA/faq.html). (Probesets or genes may be filtered by mean expression or variance (or their robust analogs such as median and median absolute deviation, MAD) since low-expressed or non-varying genes usually represent noise. Whether it is better to filter by mean expression or variance is a matter of debate; both have advantages and disadvantages, but more importantly, they tend to filter out similar sets of genes since mean and variance are usually related.). So, we ultimately chose this method.

We used data from normal samples. This decision was based on the information we obtained from reading the FAQ provided on the WGCNA official website: Data heterogeneity may affect any statistical analysis, and even more so an unsupervised one such as WGCNA. What, if any, modifications should be made to the analysis depends crucially on whether the heterogeneity (or its underlying driver) is considered "interesting" for the question the analyst is trying to answer, or not. If one is lucky, the main driver of sample differences is the treatment/condition one studies, in which case WGCNA can be applied to the data as is. At the same time, we found that the use of the normal group in WGCNA analysis is a common practice, as described in studies. such as “Weighted Gene Co-Expression Network Analysis Identifies Critical Genes in the Development of Heart Failure After Acute Myocardial Infarction.” published in Front Genet, and the “Identification of circulating hub long noncoding RNAs associated with hypertrophic cardiomyopathy using weighted correlation network analysis.” published in Mol Med Rep. in addition, the article "Characteristic Analysis of Featured Genes Associated with Cholangiocarcinoma Progression" by academician Lijuan Li also included the normal group in the cluster tree analysis. Therefore, we believe that the use of data from the normal group is reasonable. Additionally, we performed cluster tree analysis on the data and found no particularly distinct samples that needed to be removed.

Furthermore, the article from which the dataset originated (Ranjbarvaziri S, Kooiker KB, Ellenberger M, Fajardo G et al. Altered Cardiac Energetics and Mitochondrial Dysfunction in Hypertrophic Cardiomyopathy. Circulation 2021 Nov 23;144(21):1714-1731. PMID: 34672721) did not remove any group of sample data. Considering that the number of samples in our selected dataset is not large, we used all of the sample data.

Minor

The resolution of figures need to be increased. Especially in figure 4 the text is not legible.

There are a few minor english language errors that need to be corrected by the authors.

Comments on the Quality of English Language

There are a few minor english language errors that need to be corrected by the authors.

RE:

We increased the resolution and changed the layout of the images to improve the quality of the figures. In addition, we corrected the English language errors, and improved language quality.

Round 2

Reviewer 2 Report

Comments and Suggestions for Authors

I have read and reviewed your manuscript entitled " Transcriptomic and Experimental Analyses Identified lncRNA–mRNA Pair MIR210HG-BPIFC as an Immune-Related Biomarker for Hypertrophic Cardiomyopathy". Thank you for making all corrections according to my comments and I accept this manuscript in its present form.